# Teaching Back Health in the School Setting: A Systematic Review of Randomized Controlled Trials

**DOI:** 10.3390/ijerph18030979

**Published:** 2021-01-22

**Authors:** Vicente Miñana-Signes, Manuel Monfort-Pañego, Javier Valiente

**Affiliations:** Academic Unit of Physical Education, Body Languages Didactics Department, Teacher Training Faculty, University of Valencia, Av. dels Tarongers, 4, 46022 Valencia, Spain; manuel.monfort@uv.es (M.M.-P.); javivalientellin@gmail.com (J.V.)

**Keywords:** randomized controlled trial, back health, education, teaching, methodology, systematic review

## Abstract

School-based interventions have reported effectiveness on back health; however, there are no specific guidelines for teaching body mechanics and posture in primary and secondary schools. To identify, describe, and analyze the educative features of randomized controlled trials (RCTs) on back health developed to date in the school setting, a systematic review was performed following the preferred reporting items for systematic reviews and meta-analyses (PRISMA) recommendations. RCTs exclusively focused on educational setting electronic databases included PubMed, EMBASE, PEDro, Web of Science, SPORTDiscus, ERIC, and MEDLINE on the Ovid platform. Databases were searched for potentially eligible studies from the earliest date up to 18th March 2020. A total of 584 records were obtained from the database searches. A total of six articles that applied inclusion criteria were assessed for eligibility. All of these studies found improvements in postural habits and the level of knowledge with regard to back health, as well as a reduction in the prevalence of back pain. None of the studies used the student-centered method, and three studies used evaluation instruments with a pilot validation. Research on RCT interventions concerning back health in the school setting is scarce. None of the interventions applied a constructivist or student-centered method. The use of validated and standardized assessment instruments is required.

## 1. Introduction

Non-specific low back pain (LBP) is a serious and common health problem affecting a large part of the world’s population in all age groups [1], including children and adolescents [2]. A current study carried out on adolescents found an overall prevalence of low back pain of 46.7% (95% CI: 44.27 to 49.11), reporting a prevalence 42.0% (95% CI: 36.63 to 43.41) per boys and 58.0% (95% CI: 49.73 to 56.51) per girls with a statistically significant difference [3].

School-based interventions have reported effectiveness related to back health [4]. Hence, many authors have advocated for implementing back health interventions in the school setting [5] to prevent the high prevalence of back pain (BP) in children and adolescents [6], as well as the association of adolescent LBP in adulthood [7]. Among the contents that address the interventions are the improvement of knowledge, posture habits, and the practice of physical exercise [8].

According to Johnson and Deshpande [9], schools have an enormous potential to help students develop the knowledge and skills they need to be healthy. Schools are one of the leading social institutions with the responsibility of promoting health. Consequently, it seems important that back health content be more present in the school curriculum [10].

However, for a long time, there have been no specific guidelines for teaching body mechanics and posture in primary and secondary schools [11,12]. Nevertheless, many authors consider this to be important [13,14,15,16]. Rather, the studies carried out show diverse research methodologies and the use of heterogeneous evaluation tools, which do not allow the defining of a gold-standard procedure [8,17].

Most educational interventions are focused on the biomedical model used by physical therapists, or similar experts, and as treatments to prevent pain [17,18]. In addition, normally, homeroom and physical education teachers are included in the teaching process but from a traditional transmission model found in teacher-centered teaching [19]. Teachers, in addition to being specialists in the teaching area, have been shown to carry out interventions on back health in the school setting with success [5,20]. Furthermore, it is possible that teachers and students use alternative teaching methods, such as the student-centered approach, which is aligned with constructivist teaching to learning in which learners actively create, interpret, and reorganize knowledge in individual ways [21]. Hence, providing health education helps people change their attitude and behavior towards healthier lifestyles, raises awareness, and empowers and encourages people to take care of their own health [22]. This can also be related to the moral aspect in healthism where the individual is responsible for managing risks to their own health [23]. Some examples of interventions that followed these approaches used the self-management program in the school setting [24,25,26,27].

Previously, systematic reviews have been proposed concerning back health intervention in school-age children [28]. These reviews discuss the effectiveness of the interventions that deal with back health knowledge, postural habits, and exercise. However, none of them have exclusively addressed randomized controlled trials (RCTs) and focused primarily on the educational setting. In order to provide the community with the most rigorous and robust evidence in relation to what has been achieved concerning back health education interventions and how we can proceed in classrooms, this systematic review aimed to identify, describe, and analyze the educative features of RCTs on back health developed to date in the school setting. As a research question, we ask ourselves what methodologies and educative contents use school-based RCTs related to back health? Based on the literature, we hypothesized that there are few high-quality interventions (RCT), they do not present uniform research methodologies and standardized assessment instruments, and they do not delve into the pedagogical or didactical approaches.

## 2. Materials and Methods

### 2.1. Study Design

A systematic review (SR) was performed following the preferred reporting items for systematic reviews and meta-analyses (PRISMA) recommendations [29].

### 2.2. Eligibility Criteria

Randomized controlled trials (RCTs) were considered for inclusion if they took place in the school setting and involved a sample of children or adolescents (6 to 18 years old). RCTs had to involve children and adolescents with or without LBP and evaluate strategies to prevent the onset or development of BP. Outcomes had to include back health knowledge or postural habits.

### 2.3. Information Sources

Electronic databases included PubMed, EMBASE, PEDro, Web of Science, SPORTDiscus, ERIC and MEDLINE on the Ovid platform, and they were searched for potentially eligible studies from the earliest date up to 18th March 2020. Moreover, the reference lists of previous systematic reviews were checked. Articles that were not written in English or Spanish were excluded.

### 2.4. Electronic Search Strategy

An example of the search phrase with the key words and Boolean operators can be observed in Table 1. The search was applied in all fields and was limited for language reasons.

### 2.5. Study Selection

Two authors independently followed the process for selecting studies in three phases. (1) Identification: all titles and abstracts identified by the electronic search were reviewed to determine their potential relevance for the review. (2) Screening: all inclusion criteria were applied to the full text of the articles that passed the first eligibility screening. (3) Eligibility: the risk of bias in all eligible RCTs was evaluated. Disagreements were resolved by consensus, and, where necessary, by a third author (Figure 1).

### 2.6. Data Collection Process

Two authors independently extracted data from all eligible studies using standardized forms. Extracted data included the following: sample characteristics (author, setting, participant source, study groups, mean age, gender proportions, and follow-up); details regarding didactic intervention programs (e.g., teaching methodology, profession, intervention program, educative contents, duration of intervention, and assessment instruments); and baseline and follow-up outcome data (e.g., back health knowledge, postural habits, physical exercise and back pain).

To analyze the teaching methodology, we differentiate between the student-centered method and the teacher-centered method [30], as well as the biomedical model and the alternative model [18].

### 2.7. Risk of Bias in Individual Studies

The PEDro scale [31] was used to evaluate the risk of bias in all eligible RCTs. Available scores for eligible trials reported on the PEDro database were used. Only two studies were not available [32,33]. When the study was not available on the PEDro database, two previously trained authors independently scored the trial using the PEDro scale. Disagreements were resolved by Delphi consensus [34].

The authors suggested that scores of <4 are considered “poor”, 4 to 5 are considered “fair”, 6 to 8 are considered “good”, and 9 to 10 are considered “excellent” [35,36].

## 3. Results

### 3.1. Study Selection

A total of 584 records were obtained from database searches (Figure 1). Two reviewers independently excluded 567 articles after screening the titles and abstracts, and removing duplicate references. The full text was consulted when needed. After the first Delphi rounds, full agreement was reached. No additional RCTs were identified via other sources. A total of six articles that applied inclusion criteria were assessed for eligibility. Two studies were not available on the PEDro database [32,33]. Regarding the risk of bias, three studies scored 5 points on the PEDro scale, considered “fair”; two scored 6 points, considered “good”; and one scored 9, considered “excellent” (Table 2). A total of 466 articles were obtained from PubMed, followed by SPORTDiscus (*n* = 44), Medline (*n* = 38), Web of Knowledge (*n* = 27), Embase (*n* = 9), ERIC (*n* = 0), and PEDro (*n* = 0).

### 3.2. Study Characteristics

All the included studies recruited participants from a school setting according to the inclusion criteria. The sample sizes ranged from 137 to 708 students, with a mean age ranging from 8 to 12 (Table 3). One study included children from only one age group [33]. All the studies included both genders in the sample, but in one intervention, no co-education schools were included (separate schools for boys and girls) [32]. In four of the studies [13,32,33,39], the sample was assessed with a follow-up after 3 months; in the other two studies, the sample was only assessed in the pre-test and post-test [37,38] (Table 3 and Table 4).

Appendix A (Table A1) summarizes the characteristics of the didactic intervention programs of the six included studies in the systematic review [13,32,33,37,38,39]. All the studies described the educative contents. All six studies deal with conceptual contents concerning back health (Figure 2); however, these contents were only assessed in three studies [32,33,37]. Only one study performed a pilot validation of the instrument, carrying out content validity by students and experts, and reporting Cronbach’s alpha (0.84) test for reliability [32]. Regarding postural habits (Figure 3), four studies dealt with procedural contents in their programs and assessed them [13,32,37,39]. Three studies performed pilot validations of the respective evaluation instruments concerning postural habits, showing good agreement in content validity and good reliability [13,32,39]. Each study used its own tools to make assessments, and these were different for each study. None of the studies applied procedures to assess construct validity (hypothesis testing and structural validity) and criterion validity (tests regarding whether a measurement is consistent with a criterion gold standard, measured at the same time) [40] or studied error measurements using two rounds of the same test [41].

Regarding core muscles endurance (Figure 3), only two studies included procedural contents on physical exercises to improve the physical condition of the trunk muscles [37,38], but only Dullien et al. [37] assessed them. The duration of the interventions ranged from 1 h to 7 lessons (Figure 4). Only in one study [37] did teachers participate actively to administrate posture training awareness at school; however, most teachers did not write down the number of exercises as the study period progressed. The didactic approach used was the teacher-centered method [30] in all the studies. Moreover, those responsible for carrying out the theoretical and practical lessons were the members of the research groups. They used the direct instruction technique, also determined by the short time available for teaching. In some cases, mandatory physical exercises were demonstrated by the experts and carried out by the students at home [37,38].

### 3.3. Individual Studies

Most of the studies identified in this section assessed outcomes, such as knowledge about back health, postural habits, core muscles endurance, and back pain, and recorded these as an outcome (Table 4).

### 3.4. Knowledge Improvement

In the three studies that assessed knowledge about back health [32,33,37], the results indicate that only the intervention group (IG) significantly improved their knowledge from the pre- to the post-test, and in two studies, the effect remained significant 3 months after the intervention [32,33].

### 3.5. Postural Habit Improvement

All the studies registered improved postural-habit performance between the pre- and post-test, with no differences between the IG and the control group (CG) [13,37,39]. Only in the water crate-carrying task was a significant difference at post-test between the groups found [37], and in Habybabady et al. [32], differences between the IG and CG were found both in the post-test and 3 months later.

### 3.6. Core Muscles Endurance Improvement

The only RCT that assessed core muscles endurance [37] studied the effects of exercises involving push-ups, sit-ups, a balance test, standing and reaching, and hanging on wall bars. They concluded that there was no significant improvement in core muscle endurance due to the low frequency of school training.

### 3.7. Back Pain Prevention

One study reported adverse events associated with the intervention [37]. The results showed that the back pain rate increased at post-test. Only one study of the remaining five [38] reported the effectiveness of the interventions in terms of LBP prevalence. The intervention group reported significantly fewer episodes of LBP and significantly fewer lifetime first episodes of LBP compared to the control group. The other studies did not provide any more data to corroborate this [13,32,39]. In Kovacs et al. [33], no data were found on the reliability of the 8-year-old subjects’ reports on the history of LBP. Therefore, history of LBP was not gathered.

## 4. Discussion

This systematic review aimed to identify, describe, and analyze the educative features of RCTs on back health developed to date in the school setting in order to discover what methodologies and contents were used in order to teach this subject matter. In addition, with this review, we intend to provide teachers and researchers with tools to question the selected activities, assessment instruments, and the choice of the teaching method to use to address back health in schoolchildren.

In the first step of the SR, database searches (Figure 1), it can be seen that the number of RCTs concerning back health in the school setting was very low, accepting the first hypothesis. In order to obtain enough studies, Bettany-Saltikov et al. [28] decided to include prospective non-randomized studies with a control group. In the present SR, it was decided to analyze only the randomized studies to identify and provide strong evidence of “what works” in relation to educational programs and interventions [42]. RCTs include significant and rigorous process evaluations in their research designs; provide an equivalence between the intervention and the control groups, which ensures that all of the other potential factors that may influence the students’ progress are likely to be evenly distributed across the two groups; and include a consideration of the potential impact of context in relation to exploring how intervention effects could vary for different subgroups of students. Moreover, evidence confirms that it is possible to conduct RCTs in education, regardless of the nature of the educational context or of the particular type and focus of the intervention under consideration [43]. The dominant paradigm in educational research is based on qualitative methodologies (interpretive paradigm) in order to understand the multifarious interactive processes in the classroom; however, RCTs/quantitative research and qualitative research can be paired [42]. Moreover, it should be remembered that educational researchers have used the RCT procedure for a much longer period of time than therapeutic researchers [44]. This may explain the observation that more RCTs are required to be able to address a specific guideline properly.

Regarding the sample of the studies, our purpose was to inquire about all RCTs carried out in the educational field of primary and secondary education; for this reason, we covered ages between 6 and 18 years. It should be noted that all the RCTs were aimed at the primary educational stage (Table 2). However, there are also studies on younger and older ages, but they were not included because they were not RCTs. Their comparison would have been important to determine the differences in the interventions between one stage and the other. It is known that from the ages of 10 and 14, episodes of discomfort due to LBP begin to be experienced in a significant way [45,46], which may have repercussions in adulthood [7]. This makes us think that it is key to be able to intervene before, during, and after this significant peak to prevent such repercussions. On the other hand, it seems that the child–teenager stages are more suitable for learning and assimilating habits related to health and active lifestyles [16,47,48]. Moreover, regarding the advantages obtained by applying these back care intervention programs in primary schools, worth highlighting are the possibilities of offering prolonged and continuous feedback and involving a high percentage of the school population [49,50].

Four out of the six studies in the SR performed a follow-up 3 months after the intervention (Table 2). In the school setting, we need to collect data at key transition points from the same individuals over time as well as over extended periods of time. Cross sectional data collected on repeated times enables us to observe the effects of knowledge improvement, the assimilation of postural habits, and lifestyle changes. Longitudinal methods may provide a more complete procedure to research, which allows an understanding of the variation of outcomes over time [51]. As can be seen in the interventions that the improvements remained over time, although they decreased with respect to the post-test. This indicates that the interventions are effective, but that knowledge and habits about back health require a longer procedure, which could well be achieved cross-sectionally and throughout the entire stage of primary education and continued into secondary education, an aspect that we believe should be considered in the future specific guidelines.

The conceptual contents selected by the different educational programs [13,32,33,37,38,39] (Table 3) focused on the following: anatomical knowledge of the back and spine; fundamentals of LBP risk factors; the promotion of good posture; movement versus static postures; ergonomics and postural hygiene; good and bad posture while sitting; alternative sitting variations to promote dynamic sitting; healthy backpack habits (wearing, carrying, and weight); healthy lifting and carrying; back-friendly sports; nutrition; advice concerning taking responsibility for their spine; handing out the Comic Book of the Back; the promotion of physical exercise; strengthening exercises for the core muscles; and mobilization/stretching exercises to improve muscular tensions and shortenings. At this point, we believe it is important to highlight the resource proposed by Cardon et al. [16] to create 10 principles that summarize all of these theoretical contents, facilitating their teaching and learning.

The procedural contents selected by the different educational programs focused on the following: posture awareness training, such as dynamic sitting involving changing positions as being relevant; healthy lifting and carrying, which was explained by examples such as correct lifting by bending the knees for the consistent distribution of weight; standing; and pushing and pulling demonstrations. Concerning core muscle endurance [37], static and dynamic exercises involved the following: the plank, the crunch, hip lifts, flexion of the back muscles, ball exercises, balance, breathing and relaxation [13,39], and joint mobility exercises, emphasizing moving the back through the full range of specific adherence-enhancing strategies [38]. The most complete educational content program could be found in the study by Dullien et al. [37]. There, the contents related to the concepts, the practice of postural habits, and physical exercises developed for the improvement of core muscle endurance, accompanied by an appendix, are described in detail. We see a fairly complete selection of content amongst all of the programs, although the resources of education are much broader, and back health can be addressed in many more ways. For example, one can make use of body expression or dramatic play [16]; sports; sensory-perceptual abilities, such as postural attitude, postural scheme, and balance; and a multitude of educational games that integrate the contents of back health [5]. For these reasons, we believe that the specific guideline that can be created in the next future should contemplate all of these educational contents and propose a sequencing of them throughout the educational stage of primary and secondary education.

With regard to the assessment tools, only one study used an instrument with pilot validity and reliability to evaluate knowledge [32]. Regarding postural habits, three studies performed pilot validations [13,32,39]. To the best of our knowledge, there are few assessment instruments concerning knowledge and postural habits that have been validated and published with their respective psychometric analyses and discussions [52,53,54,55]. If researchers agree to use a single set of postural metrics, it might provide clearer evidence about which treatments work best and why [17]. Therefore, this confirms the second hypothesis of the study that stated that intervention programs do not present uniform research methodologies and standardized assessment instruments. In this sense, a specific guideline could determine a reference methodology that standardizes the intervention protocols in the educational field, allowing some flexibility to be adapted to the context of each case.

If RCT interventions on back health in the school setting, with a duration ranging between one hour and seven sessions, were found to be effective in improving knowledge, postural habits, and reducing LBP but with losses observed in the follow-up, it might be understood that educative interventions on back health should be longitudinal and compulsory in the curriculums during primary school.

The intervention programs included theoretical and practical classes, with the exception of the study by Kovacs et al. [33], which only handed out a didactic resource to the children, the Comic Book of the Back, and did not apply teaching methodology to address this. Moreover, those responsible for giving the lessons were the members of the research groups. It is indicated that the researchers came to the educational centers for between four to seven sessions to offer theoretical and practical explanations, respectively [13,32,37,38,39]. Some studies specify the use of theoretical posters [37] and theoretical pamphlets [32], while others detail the order of the sessions, with specific timings and the activities the students had to do. However, due to the short time available and the non-intervention of the teachers, the classes had to follow a teacher-centered teaching style or method (TCM). In some cases, mandatory physical exercises were shown and demonstrated by the experts for the students to carry out at home [37,38]. This confirms the third hypothesis that referred to the fact that the studies did not delve into the explanation and details of the pedagogical or didactic approaches used. In a quasi-experimental study on back health, the authors indicated the use of a methodology based on student-centered teaching (SCT) through guided discovery and active hands-on methods, such as games and dramatic play [16]. Should the interventions include education professionals who are knowledgeable about the different methodologies? Would it make sense for work teams to be multidisciplinary? Only one study [37] was made up of a multidisciplinary team, and the teachers participated actively to administrate posture training awareness at school; however, most teachers did not write down the number of exercises as the study period progressed.

Constructivist and student-centered approaches to pedagogy have spawned a wide variety of different active learning modes, strategies, and techniques to carry out contents. For example, as learning modes, we could highlight cooperative learning, collaborative learning, problem-based or inquiry learning, service learning, experiential learning, participant learning, etc., with most of them being developed in the school setting [30]. These learning modes require varied-group organization, group learning, small group work, peer-assisted learning, peer-tutoring, workshops, educative corners, internships, community service, field work, or outside-of-class collaboration. Strategies such as cooperating in order to attain a common goal, mutually relying upon the resources provided by different members of the group, and engaging in face-to-face interaction will be key to the development of these approaches [30]. In addition, analysis, synthesis, interdisciplinary exploration, self-observation, evaluation, the ultimate application of information, and reflection techniques are used.

It seems that despite reform initiatives, traditional teaching still dominates education, with instructors or teachers mainly considering themselves as transmitters of knowledge with traditional beliefs about teaching and learning [56]. Nowadays, students are expected to be prepared with competencies empowering them to think analytically and critically, solve realistic problems, reflect on what they know, work in collaboration with others, and manage their own learning [57]. In student-centered learning (SCL), the students are placed at the center of teaching and learning, taking an active role in their own learning as contrary to the high level of teacher control and educational content found in TCM [19]. SCT focuses on how students learn instead of how teachers teach. In a learner-centered classroom, teachers abandon master classes, lecture notes, lecture-based classrooms, and power point presentations for a more dynamic, participatory, engaging, collaborative style of teaching [58]. Teachers may act as facilitators, guides, navigators, and co-learners, encouraging students to take responsibility for learning while modelling learning processes and providing opportunities to develop their learning skills. For reform to be successful, it needs to be systemic, simultaneously addressing all interdependent components of the educational system, most importantly the curriculum; professional development and school culture-revisions carried out on only one component would result in failure unless concomitant changes are made to all of the other components [59].

Only in two studies [13,38] were teachers familiarized with the study and data collection procedures in a short information session prior to the school visit. In order to improve the interventions and the knowledge of the students, over the last two decades, a growing body of international research in teacher education has focused on the significance, effectiveness, and chances associated with learning communities (LC) [60] and communities of practice [61]. Along the same lines, it also seems important to identify whether the interventions have taken into account teacher training and the type of training that has been carried out. Moreover, it is important to know whether the intervention program has established the basic principles for implementing the intervention program and if there is any loyalty procedure for the intervention in order to standardize it and explain how the educational activities and methodologies are taught. Research on teaching and teacher education has improperly ignored research questions dealing with the content of taught lessons [62]. For these reasons, Shulman introduced a new concept called pedagogical content knowledge (PCK), which refers to teachers’ interpretations and transformations of subject matter knowledge in the context of facilitating student learning. Understanding more about teachers as learners, what they need to know, and how they learn their craft can help to clarify the role of formal teacher training in regard to learning in order to teach [63]. Balague et al. [20] proposed that teachers must be trained for the correct development of back health programs in the school setting. Teachers are a key agent in this matter. A trained teacher has the ability to promote student learning of physical literacy, because students spend many hours sitting during the school day [64].

Finally, it should be noted that no interventions based on the comprehensive model have been found. The determinants of health indicate that in order to bring about changes in people’s health, it is necessary to carry out comprehensive interventions that take into account interdisciplinary and multidisciplinary facets [65]. Health should not be approached in isolation, but rather, it depends on many agents in society, as advocated by the socioecological model [66].

Starting from these RCT interventions, based mainly on the biomedical model, it is important that the educational community join this field of study to complement it, since it has a direct application in their field of work, as suggested by several other authors [18,20]. The studies are based on the prejudice of presenting students as “sick” or “future sick”. This medical viewpoint conditions the orientation of the work conducted with the students and does not allow the student to be introduced to the complex learning process. For example, the study by Hill and Keating [38] presents, from an educational point of view, a very reductionist view of the intervention in which it only evaluates the evolution of back pain. Thus, the educational purpose is not to avoid diseases exclusively, but to better understand our body to adapt to different situations and thus be able to achieve the highest possible level of well-being.

As future lines of research, prior training of teachers who apply the interventions and the follow-up is required. In addition, the contents of back health from a more constructivist methodology should be applied, in which the student is the protagonist of the teaching-learning process. Moreover, the standardized development and selection of a set of validated and reliable evaluation instruments is required. Subsequently, the development of specific guidelines for the teaching of back health in primary and secondary education would be of great help to educational communities.

Regarding implications for school health, clearly, this systematic review aims to be a valuable tool for the entire educational community, and especially for physical education teachers, to better understand how to address back health education in the school setting. In addition to the selection of randomized controlled trials, teachers will be able to review the summary and selection of contents and designs of the intervention programs carried out. On the other hand, the teaching community is encouraged to implement postural education in schools.

## 5. Conclusions

Research concerning RCT interventions on back health in the school setting is scarce. The main educational contents on back health seem to have been identified; however, none of the interventions applied a constructivist or student-centered method. The use of validated and standardized assessment instruments is required. Teachers should consider back health as a transversal content, approaching it over the years from a comprehensive model. We suggest developing interventions related to back health at school age to improve the pedagogical content of knowledge so that the educational field, and its professionals, can help us improve its application in field work.

## Figures and Tables

**Figure 1 ijerph-18-00979-f001:**
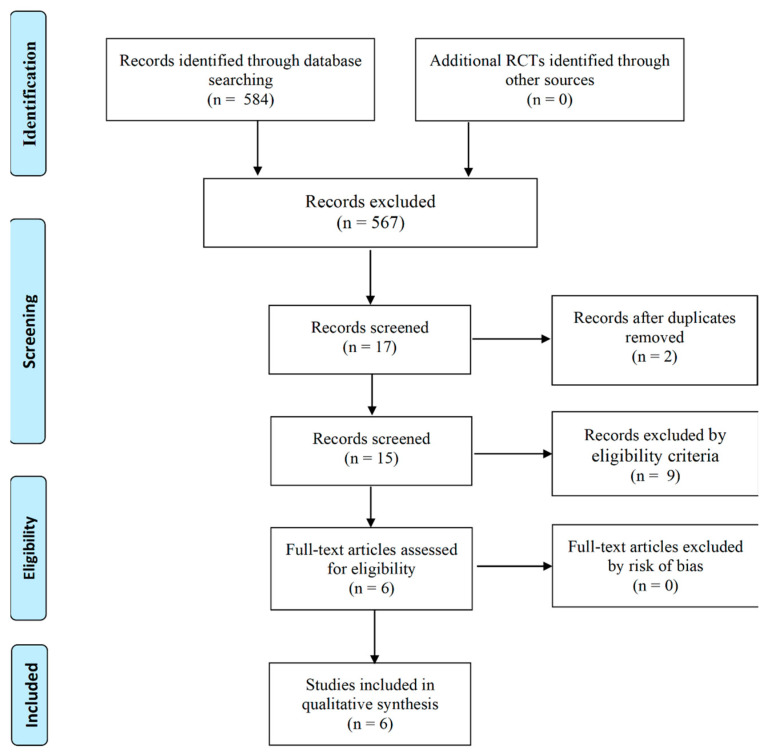
Flow diagram of search strategy.

**Figure 2 ijerph-18-00979-f002:**
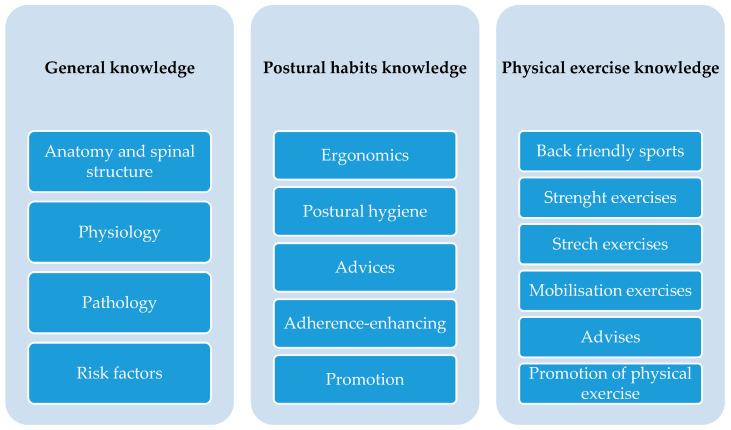
Types of conceptual contents developed in RCT interventions for back health in the school setting.

**Figure 3 ijerph-18-00979-f003:**
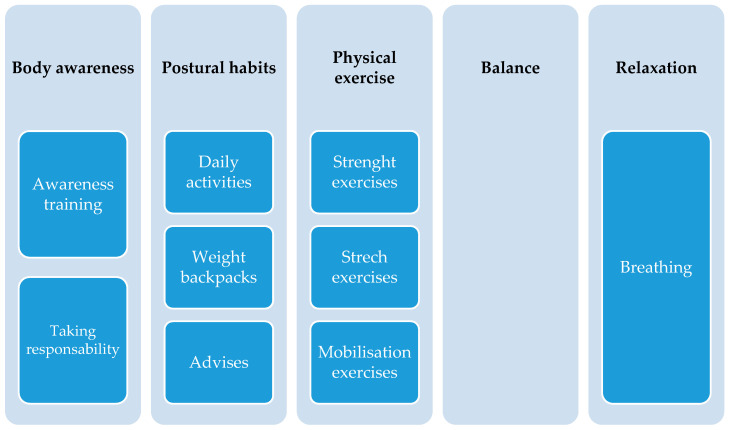
Types of procedural contents developed in RCT interventions for back health in the school setting.

**Figure 4 ijerph-18-00979-f004:**
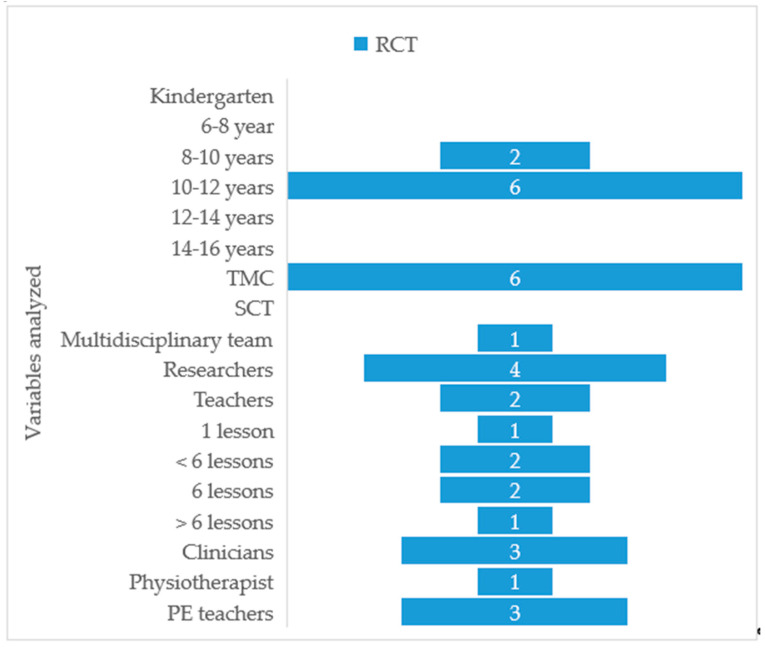
Summary of variables of the intervention programs developed in RCTs for back health in the school setting. RCT: randomized controlled trial; TMC: teacher-centered teaching style or method; SCT: student-centered teaching; PE teachers: physical education teachers.

**Table 1 ijerph-18-00979-t001:** Electronic search strategy.

Bolean Operators
Education or Elementary Education or Primary School or High School or Secondary School
Child * or adol * or student or Young or youth
Back pain or low back pain or back health
Randomized Controlled Trial
(#1) and (#2) and (#3) and (#4) and (#5)

Complete phrase: (Education or elementary education or primary school or high school or secondary school) and (Child * OR adol * or student or Young or youth) and (Back pain or low back pain or back health) and (Randomized Controlled Trial).

**Table 2 ijerph-18-00979-t002:** Risk of bias in individual randomized controlled trials (RCTs).

RCT	Eligibility Criteria	Random Allocation	Concealed Allocation	Baseline Comparability	Blind Subjects	Blind Therapists	Blind Assessors	Adequate Follow-up	Intention-to-Treat Analysis	Between-Group Comparisons	Point Estimates and Variability	PEDro Scores (10)
[37]	0	1	0	1	0	0	1	0	0	1	1	5
[32]	1	1	0	1	0	0	0	1	1	1	1	6
[38]	1	1	1	1	0	0	0	0	1	1	1	6
[33]	1	1	1	1	1	0	1	1	1	1	1	9
[13]	1	1	0	1	0	0	0	1	0	1	1	5
[39]	1	1	0	1	0	0	0	1	0	1	1	5

aPEDro score. Each satisfied item (except the first item) contributes 1 point to the total PEDro score (range = 0–10 points).

**Table 3 ijerph-18-00979-t003:** Sample characteristics RCTs.

Author and Funding	Setting	Participants	Groups	Mean Age (SD)	Gender	Follow-Up
Clinicians and sport teachers, funding of EUR 10,000 [37]	Two schools or German “gymnasiums”	*n* = 176 10–12 y 5th grade	4 classes IG (*n* = 96); 4 classes CG (*n* = 86); cluster-randomized	IG: 10.6 ± 0.44; CG: 10.5 ± 0.43	100 G; 76 B	-
Clinicians with funding [32]	Elementary school children; no co-education schools in Iran; Iran’s south-eastern city of Zahedan	*n* = 404 10–11 y 5th grade	IG (*n* = 203); CG (*n* = 201); 25–30 students/classroom; 4 schools each of boys and girls; systematic random sampling method from each list	-	IG: 101 G, 102 B; CG: 104 G, 97 B	3 months
Physiotherapy [38]	Seven primary schools; North Shore, City district, Auckland North region, New Zealand; 2011 academic year	*n* = 708 8–11 y	4 schools with IG (*n* = 469); 4 schools with CG (*n* = 239)	IG: 9.4 ±0.63; CG: 9.3 ±0.64	IG: 48%; CG: 49%	-
Clinicians with funding [33]	Twelve schools, (six public, four concerted, and two private) in Majorca, Spain	*n* = 456 8 y	6 schools for IG (*n* = 266, 53.5%); 6 schools for CG (*n* = 231, 46.5%)	-	IG: 121 G (45.5%); CG: 116 G (50.2%)	3 months
Physical education university teachers [13]	Six classes from two primary schools in Majorca, Spain	*n* = 137 10–12 y 5th–6th grades	3 classes IG (*n* = 63); 3 classes CG (*n* = 74)	10.72 ± 0.672	IG: 33 B, 30 G; CG: 38 B 36 G	3 months
Physical education university teachers [39]	Six classes from two primary schools in Majorca, Spain	*n* = 137 10–12 y 5th–6th grades	3 classes IG (*n* = 63); 3 classes CG (*n* = 74)	10.72 ± 0.672	IG: 33 B, 30 G; CG: 38 B 36 G	3 months

Y: year; G: girls; B: boys; IG: intervention group; CG: control group

**Table 4 ijerph-18-00979-t004:** Baseline and follow-up outcome data.

Study	Knowledge			Postural Habits			Physical Exercise			Back Pain		
	BL	ST	IT	BL	ST	IT	BL	ST	IT	BL	ST	IT
[37]	IG: 14.42 ± 3.03; CG: 14.80 ±5.05	IG: 17.17 ± 2.84 *; CG: 14.57 ± 4.42	-	IG: 5.7 ± 1.9 *; CG: 6.1 ±1.7	IG: 8.2 ±2.0 *; CG: 7.7 ±2.1	-	Sit-ups, IG: 20.52 ± 4.55; CG: 18.29 ± 4.42	Sit-ups, IG: 20.00 ± 4.89; CG: 19.64 ± 4.69	-	IG: 26.7%; CG: 24.4%	IG: 52.6%; CG: 68.4%	-
[32]	IG: 43.4 ± 12.93; CG: 47.0 ± 12.76	IG: 74.5 ± 19.60 *; CG: 48.1 ± 13.78	IG: 60.5 ± 24.32 *; CG: 39.6 ± 15.89	IG: 53.3 ± 16.34; CG: 54.7 ± 13.57	IG: 75.8 ± 18.58 *; CG: 56.0 ± 16.43	IG: 65.5 ± 20.34 *; CG: 49.2 ± 14.37	-	-	-	LBP, 18.3% (*n* = 57)	-	-
[38]	-	-	-	-	-	-	-	-	-	LBP, IG: 46% (*n* = 218); CG: 47% (*n* = 112)	LBP, IG: 16% (*n* = 58); CG: 24% (*n* = 43)	-
[33]	IG: 7.0 (IQR 6; 8); CG: 8.0 (IQR 7; 9)	IG: 9 (IQR 8; 9) *; CG: 8 (IQR 7; 9)	IG: 9 (IQR 8; 10) *; CG: 9 (IQR 8; 9)	-	-	-	-	-	-	-	-	-
[13]	-	-	-	IG: 3.4; CG: 3,7	IG: 4.2 *; CG: 3.9	IG: 4.0 *; CG: 3.6	-	-	-	LBP, IG: 28.6% (*n* = 18); CG: 32.4% (*n* = 24)	-	-
[39]	-	-	-	IG: 2.174 ± 0.959; CG: 2.581 ± 0.811	IG: 2.7 *; CG: 2.45	IG: 2.62 *; CG: 2.5	-	-	-	LBP, IG: 28.6% (*n* = 18); CG: 32.4% (*n* = 24)	-	-

BL: baseline; ST: short term; IT: intermediate term; * significant differences; LBP: low back pain; IQR: interquartile ranges.

## Data Availability

The data reported in the study are explained in the methodology.

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
