# Peer review of "Teaching Back Health in the School Setting: A Systematic Review of Randomized Controlled Trials"

_ijerph, 2021, doi:10.3390/ijerph18030979_

Round 1
Reviewer 1 Report
Dear Authors, thank you for the paper. Please find some comments and questions below.
1- The main aim of the study wrote based on there are no specific guidelines for teaching body mechanics and posture in primary and secondary.
Please provide a refrences in this point and explain why there are no guidelines as well as why it should be in primary and secondary!
2- In P.1 line 36 ''seems important that back health content be more present in the school curriculum (7).
- What does it mean ''seems'' exactly?
- Is it neglected?
- 3- Do you think that the experts who made the education contents needs to draw their attention to that point?
3- Is the educational contents the same between diffrent cities or not?
- some of the presented references discuss the the effects of a postural education program...can the authors indicate the differences between the the effects of a postural education program and guidelines for teaching body mechanics and posture? The children in early stage learn through the movements more than theoretical education!
4- Line 64 we hypothesized that there are few high-quality interventions. Can you please add some parameter describe this ''quality''
5- The important results of this study were presented in a way that was not 100% interested for the readers. Can you revisit the key results for presentation in multiple graphs? It will leave draw the readers' attention and make following easier.
Author Response
Responses to Reviewers' Comments
[IJERPH] Manuscript ID: ijerph-1043243
We thank the reviewers for their thoughtful and in-depth comments concerning our manuscript. Your suggestions helped us improve the quality of our paper. We carefully considered every comment and made the appropriate changes, which are highlighted in red font. Besides the manuscript has been edited by a specialist native speaker. Our point-by-point responses are also noted below.
Reviewer #1:
Dear Authors, thank you for the paper. Please find some comments and questions below.
1- The main aim of the study wrote based on there are no specific guidelines for teaching body mechanics and posture in primary and secondary.
Please provide a refrences in this point and explain why there are no guidelines as well as why it should be in primary and secondary!
Author’s response:
In line 214 we have included the next sentence:
Perhaps, this explains that more RCTs are required to be able to address a specific guideline properly.
Besides, in the discussion section we explained why important primary and secondary education is. Find it at line 215:
“Regarding the sample of the studies, it should be noted that all the RCTs were aimed at the primary educational stage (Table 2). This is because it is known that from the ages of 10 and 14 episodes of discomfort due to low back pain begin to be experienced in a significant way (42, 43), which may have repercussions in adulthood (4). On the other hand, it seems that the child- teenager stages are more suitable for learning and assimilating habits related to health and active lifestyles (13, 44, 45). Moreover, regarding the advantages obtained by applying these back care intervention programs in primary schools, worth highlighting is the possibility of offering prolonged and continuous feedback, and involving a high percentage of the school population (46, 47).”
Moreover, in line 233 we have included the next sentence:
“This indicates that the interventions are effective, but that knowledge and habits about back health require a longer procedure which could well be done cross-sectionally and throughout the entire stage of primary education and continued into secondary education, an aspect that we believe should be considered in the future specific guidelines.”
And in line 259 we have included the next paragraph:
“For these reasons, we think that the specific guideline that can be created in the next future should contemplate all these educational contents and propose a sequencing of them throughout the educational stage of primary and secondary education”.
In line 270 we have included also the next paragraph:
“In this sense, a specific guideline could determine a reference methodology that standardizes the intervention protocols in the educational field, allowing some flexibility to be adapted to the context of each case.”
In line 274 we explain how should be future guidelines “If RCT interventions on back health in the school setting, with a duration ranging between 1 hour and 7 sessions, have been found to be effective in improving knowledge, postural habits, and reducing low back pain, but in the follow up losses were observed, it might be understood that educative interventions on back health should be longitudinal and compulsory in the curriculums during primary and secondary school”
2- In P.1 line 36 ''seems important that back health content be more present in the school curriculum (7).
- What does it mean ''seems'' exactly?
- Is it neglected?
- 3- Do you think that the experts who made the education contents needs to draw their attention to that point?
Author’s response: Actually, the educational contents refer to health care and postural habits, rather what we think is a priority is to disseminate a specific guideline that explains how to carry out back care from a socio-ecological, more global and comprehensive model.
Moreover, in line 341 we explain that “Balague et al. (17) proposed that teachers must be trained for the correct development of back health programs in the school setting. Teachers are a key agent in this matter. A trained teacher has the ability to promote student learning of physical literacy, because students spend many hours sitting during the school day (61).”
Line 345: Finally, it should be noted that no interventions based on the comprehensive model have been found. The determinants of health indicate that in order to bring about changes in people's health, it is necessary to carry out comprehensive interventions that take into account interdisciplinary and multidisciplinary facets (62). Health should not be approached in isolation, but rather it depends on many agents in society, as advocated by the socio-ecological model (63).
3- Is the educational contents the same between diffrent cities or not?
- some of the presented references discuss the the effects of a postural education program...can the authors indicate the differences between the the effects of a postural education program and guidelines for teaching body mechanics and posture? The children in early stage learn through the movements more than theoretical education!
Author’s response:
4- Line 64 we hypothesized that there are few high-quality interventions. Can you please add some parameter describe this ''quality''.
Author’s response: Yes, we talk about RCT, so we have updated the sentence: “few high-quality interventions (RCT)”
5- The important results of this study were presented in a way that was not 100% interested for the readers. Can you revisit the key results for presentation in multiple graphs? It will leave draw the readers' attention and make following easier.
Author’s response: We have incorporated some new graphs to focus and make easier the reading and understanding of the main results.
Figure 2. Types of conceptual contents developed in RCT interventions for back health in the school setting.
Figure 3. Types of procedural contents developed in RCT interventions for back health in the school setting.
RCT: Randomized Controlled Trial; TMC: teacher-centered teaching style or method; SCT: student-centered teaching; PE teachers: Physical Education teachers.
Figure 4. Summary of variables of the intervention programs developed in RCTs for back health in the school setting.
Reviewer 2 Report
Peer review IJERPH (ISSN 1660-4601)
Manuscript ID: ijerph-1043243
Teaching back health in the school setting: a systematic review of randomized control trials
This is a systematic review to identify, describe and analyze the educative features of RCTs on back health developed to date in the school setting. The authors included 6 articles. All of these studies found improvements in postural habits and the level of knowledge with regard to back health, as well as a reduction in the prevalence of back pain.
Overall, this is a well-done systematic review raising an important discussion.
Major comments:
- I miss an overview of the problem in the introduction. What is the prevalence of back pain among schoolchildren (primary and secondary school children)? “How big is the problem?”
- The aim is described as: ‘This systematic review aimed to identify, describe and analyze the educative features of RCTs on back health developed to date in the school setting. As a research question, we ask ourselves what methodologies and educative contents use school-based RCTs related to back health?’
- Framing the research question the way the authors did; it seems very descriptive. Especially, because the authors indicate that previous systematic reviews have addressed the effectiveness of interventions with deal with back health knowledge, postural habits, and exercise. Do the authors focus on describing the interventions in RCTs, or on the methodology of the RCTs?
- The knowledge gap should be more obvious, what do we learn from this systematic review if school-based interventions have already reported effectiveness on back health. Why is the educational setting (and what does this entail) so important?
- Children 6-18 years of age is a large variation in age. Do you expect/Did you find any differences in age groups? The authors briefly touched upon this issue in the discussion, but I would like the authors to discuss this in more detail. The studies in this systematic review, only included children 8-12 years of age. Why do you think this is the case? Conclusions drawn upon this study can only be generalized to children 8-12 years of age. The effects of these types of interventions might be different for older adolescents.
- The authors should specify the important outcomes of interest to the review, and define acceptable ways of measuring them.
- The risk of bias assessment of the included studies should be included in the results section.
- Your conclusion: “If RCT interventions on back health in the school setting, with a duration ranging between 1 hour and 7 sessions, have been found to be effective in improving knowledge, postural habits, and reducing low back pain, but in the follow up losses were observed, it might be understood that educative interventions on back health should be longitudinal and compulsory in the curriculums during primary and secondary school.” Is too bold, and cannot be stated like this based on your results. The included studies did not include any secondary school children (at least not older than 12 years of age).
Minor comments:
- Change Randomized control trials to ‘Randomized Controlled Trials’
- I would recommend the authors to have the manuscript checked by a native English speaker or a translator, or give the manuscript another careful read-through. I found non-academic language, an example: “Page 2, line 57: These talk about…”
- Table 4: What are the effect sizes reported in the table? Betas? Can you please add a legenda to this table. Please describe this in the text as well.
Author Response
Responses to Reviewers' Comments
[IJERPH] Manuscript ID: ijerph-1043243
We thank the reviewers for their thoughtful and in-depth comments concerning our manuscript. Your suggestions helped us improve the quality of our paper. We carefully considered every comment and made the appropriate changes, which are highlighted in red font. Besides the manuscript has been edited by a specialist native speaker. Our point-by-point responses are also noted below.
Reviewer #2:
Teaching back health in the school setting: a systematic review of randomized control trials
This is a systematic review to identify, describe and analyze the educative features of RCTs on back health developed to date in the school setting. The authors included 6 articles. All of these studies found improvements in postural habits and the level of knowledge with regard to back health, as well as a reduction in the prevalence of back pain.
Overall, this is a well-done systematic review raising an important discussion.
Major comments:
- I miss an overview of the problem in the introduction. What is the prevalence of back pain among schoolchildren (primary and secondary school children)? “How big is the problem?”
Author’s response: Now, the first paragraph of the introduction section talk about the problem and reference the current studies on the topic.
Non-specific low back pain (LBP) is a serious and common health problem affecting a large part of the world's population in all age groups (1), including children and adolescents (2). A current study carried out in adolescents found an overall prevalence of low back pain of 46.7% (95% CI: 44.27 to 49.11), reporting a prevalence 42.0% (95% CI: 36.63 to 43.41) per boys and 58.0% (95% CI: 49.73 to 56.51) per girls with a statistically significant difference (3).
- The aim is described as: ‘This systematic review aimed to identify, describe and analyze the educative features of RCTs on back health developed to date in the school setting. As a research question, we ask ourselves what methodologies and educative contents use school-based RCTs related to back health?’
- Framing the research question the way the authors did; it seems very descriptive. Especially, because the authors indicate that previous systematic reviews have addressed the effectiveness of interventions with deal with back health knowledge, postural habits, and exercise. Do the authors focus on describing the interventions in RCTs, or on the methodology of the RCTs?
Author’s response: Finally, the authors focus on describing the interventions in RCTs.
For theses reasons we have change the next sentence:
“In order to provide the community with the most rigorous and robust evidence in relation to what has been done concerning back health education interventions, and how can we proceed in the classrooms”
In addition to finding the results of the analysis of the interventions by tables, in the discussion we have tried to discuss the characteristics of these interventions.
The conceptual contents selected by the different educational programs (13, 32, 33, 37-39) (Table 3) focused on: anatomical knowledge of the back and spine; fundamentals of LBP risk factors; the promotion of good posture; movement versus static postures; ergonomics and postural hygiene; good and bad posture while sitting; alternative sitting variations to promote dynamic sitting; healthy backpack habits (wearing, carrying and weight); healthy lifting and carrying; back-friendly sports; nutrition; advice concerning taking responsibility for their spine; handing out a Comic Book of the Back; the promotion of physical exercise; strengthening exercises for the core muscles, and mobilization/ stretching exercises to improve muscular tensions and shortenings. At this point, we believe it is important to highlight the resource proposed by Cardon et al. (16) to create 10 principles which summarize all these theoretical contents, facilitating their teaching and learning.
The procedural contents selected by the different educational programs focused on: posture awareness training, such as dynamic sitting involving changing positions as being relevant; healthy lifting and carrying, which was explained by examples such as correct lifting by bending the knees for the consistent distribution of weight; standing; pushing and pulling demonstrations. Concerning core muscles endurance (37), static and dynamic exercises involved: the plank, the crunch, hip lifts, flexion of the back muscles, ball-exercises, balance, breathing and relaxation (13, 39); and joint mobility exercises, emphasizing moving the back through the full range of specific adherence-enhancing strategies (38). The most complete educational content program could be found in the study by Dullien et al (37). There, the contents related to the concepts, the practice of postural habits and physical exercises developed for the improvement of core muscle endurance, accompanied by an appendix, are described in detail. We see a fairly complete selection of content amongst all the programs, although the resources of education are much broader, and back health can be addressed in many more ways. For example, you can make use of body expression or dramatic play (16), sports, sensory-perceptual abilities such as postural attitude, postural scheme, and balance, as well as a multitude of educational games that integrate the contents of back health (5).
With regard to the assessment tools, only one study used an instrument with pilot validity and reliability to evaluate knowledge (32). Regarding postural habits, three studies performed pilot validations (13, 32, 39). To the best of our knowledge, there are few assessment instruments concerning knowledge and postural habits which have been validated and published with their respective psychometric analyses and discussions (52-55). If researchers agree to use a single set of postural metrics, it might provide clearer evidence about which treatments work best, and why (17). Therefore, this confirms the second hypothesis of the study which stated that intervention programs do not present uniform research methodologies and standardized assessment instruments.
If RCT interventions on back health in the school setting, with a duration ranging between 1 hour and 7 sessions, have been found to be effective in improving knowledge, postural habits, and reducing LBP, but in the follow up losses were observed, it might be understood that educative interventions on back health should be longitudinal and compulsory in the curriculums during primary and secondary school.
- The knowledge gap should be more obvious, what do we learn from this systematic review if school-based interventions have already reported effectiveness on back health. Why is the educational setting (and what does this entail) so important?
Author’s response: This is a very interesting and key question. Throughout the article we have tried to explain the following:
In the introduction section we talked about:
School-based interventions have reported effectiveness related to back health (4). Hence, many authors have advocated for implementing back health interventions in the school setting (5) to prevent the high prevalence of back pain (BP) in children and adolescents (6), as well as the association of adolescent LBP in adulthood (7).
According to Johnson and Deshpande (9), schools have an enormous potential to help students develop the knowledge and skills they need to be healthy. Schools are one of the leading social institutions with the responsibility of promoting health. Consequently, it seems important that back health content be more present in the school curriculum (10).
On the other side, we discussed the next contect:
Regarding the sample of the studies, it should be noted that all the RCTs were aimed at the primary educational stage (Table 2). This is because it is known that from the ages of 10 and 14 episodes of discomfort due to LBP begin to be experienced in a significant way (45, 46), which may have repercussions in adulthood (7). On the other hand, it seems that the child- teenager stages are more suitable for learning and assimilating habits related to health and active lifestyles (16, 47, 48). Moreover, regarding the advantages obtained by applying these back care intervention programs in primary schools, worth highlighting is the possibility of offering prolonged and continuous feedback, and involving a high percentage of the school population (49, 50)
- Children 6-18 years of age is a large variation in age. Do you expect/Did you find any differences in age groups? The authors briefly touched upon this issue in the discussion, but I would like the authors to discuss this in more detail. The studies in this systematic review, only included children 8-12 years of age. Why do you think this is the case? Conclusions drawn upon this study can only be generalized to children 8-12 years of age. The effects of these types of interventions might be different for older adolescents.
Author’s response: You refocus the issue on another key point in this line of research. We hope to continue investigating to try to clarify interventions sequenced by courses and years and to approach back education with a good guideline.
We have tried to improve the paragraph that refers to the ages to try to make it clearer. We hope it will be better.
Regarding the sample of the studies, our purpose was to inquire about all RCTs carried out in the educational field of primary and secondary education, for this reason we covered ages between 6 and 18 years. It should be noted that all the RCTs were aimed at the primary educational stage (Table 2). However, there are also studies at younger and older ages, but they were not included because they were not RCT. Their comparison would have been important to determine the differences in the interventions between one stage and the other. It is known that from the ages of 10 and 14 episodes of discomfort due to LBP begin to be experienced in a significant way (45, 46), which may have repercussions in adulthood (7). This makes us think that it is key to be able to intervene before, during and after this significant peak to prevent it. On the other hand, it seems that the child- teenager stages are more suitable for learning and assimilating habits related to health and active lifestyles (16, 47, 48). Moreover, regarding the advantages obtained by applying these back care intervention programs in primary schools, worth highlighting is the possibility of offering prolonged and continuous feedback, and involving a high percentage of the school population (49, 50).
- The authors should specify the important outcomes of interest to the review, and define acceptable ways of measuring them.
Author’s response: We have incorporated some new graphs to focus and make easier the reading and understanding of the main results.
Figure 2. Types of conceptual contents developed in RCT interventions for back health in the school setting.
Figure 3. Types of procedural contents developed in RCT interventions for back health in the school setting.
RCT: Randomized Controlled Trial; TMC: teacher-centered teaching style or method; SCT: student-centered teaching; PE teachers: Physical Education teachers.
Figure 4. Summary of variables of the intervention programs developed in RCTs for back health in the school setting.
Regarding the acceptable ways of measuring them we introduce valid and reliable evaluation instruments developed in school setting:
With regard to the assessment tools, only one study used an instrument with pilot validity and reliability to evaluate knowledge (32). Regarding postural habits, three studies performed pilot validations (13, 32, 39). To the best of our knowledge, there are few assessment instruments concerning knowledge and postural habits which have been validated and published with their respective psychometric analyses and discussions (52-55). If researchers agree to use a single set of postural metrics, it might provide clearer evidence about which treatments work best, and why (17). Therefore, this confirms the second hypothesis of the study which stated that intervention programs do not present uniform research methodologies and standardized assessment instruments. In this sense, a specific guideline could determine a reference methodology that standardizes the intervention protocols in the educational field, allowing some flexibility to be adapted to the context of each case.
- The risk of bias assessment of the included studies should be included in the results section.
Author’s response: About the Risk of bias assessment in the results section
Results
3.1. Study selection
A total of 584 records were obtained from database searches (Figure 1). Two reviewers independently excluded 567 articles after screening the titles and abstracts, and removing duplicate references. The full text was consulted when needed. After the first Delphi rounds, full agreement was reached. No additional RCTs were identified via other sources. A total of 6 articles that applied inclusion criteria were assessed for eligibility. Two studies were not available on the PEDro database (32, 33). Regarding the risk of bias, three studies scored 5 points on the PEDro scale, considered “fair”, two scored 6 points, considered “good”, and one scored 9, considered “excellent” (Table 2). 466 articles were obtained from Pubmed, followed by SPORTDiscus (n = 44), Medline (n = 38), Web of Knowledge (n = 27), Embase (n = 9), ERIC (n = 0) and PEDro (n = 0).
Table 2. Risk of bias in individual RCTs.
|
RCT |
Eligibility criteria |
Random allocation |
Concealed allocation |
Baseline comparability |
Blind subjects |
Blind therapists |
Blind assessors |
Adequate follow-up |
Intention-to-treat analysis |
Between-group comparisons |
Point estimates And variability |
PEDro scorea (/ 10) |
|
(37) |
0 |
1 |
0 |
1 |
0 |
0 |
1 |
0 |
0 |
1 |
1 |
5 |
|
(32) |
1 |
1 |
0 |
1 |
0 |
0 |
0 |
1 |
1 |
1 |
1 |
6 |
|
(38) |
1 |
1 |
1 |
1 |
0 |
0 |
0 |
0 |
1 |
1 |
1 |
6 |
|
(33) |
1 |
1 |
1 |
1 |
1 |
0 |
1 |
1 |
1 |
1 |
1 |
9 |
|
(13) |
1 |
1 |
0 |
1 |
0 |
0 |
0 |
1 |
0 |
1 |
1 |
5 |
|
(39) |
1 |
1 |
0 |
1 |
0 |
0 |
0 |
1 |
0 |
1 |
1 |
5 |
aPEDro score. Each satisfied item (except the first item) contributes 1 point to the total PEDro score (range=0–10 points).
- Your conclusion: “If RCT interventions on back health in the school setting, with a duration ranging between 1 hour and 7 sessions, have been found to be effective in improving knowledge, postural habits, and reducing low back pain, but in the follow up losses were observed, it might be understood that educative interventions on back health should be longitudinal and compulsory in the curriculums during primary and secondary school.” Is too bold, and cannot be stated like this based on your results. The included studies did not include any secondary school children (at least not older than 12 years of age).
Author’s response: Totally agree. He have change the conclusion like following:
If RCT interventions on back health in the school setting, with a duration ranging between 1 hour and 7 sessions, have been found to be effective in improving knowledge, postural habits, and reducing LBP, but in the follow up losses were observed, it might be understood that educative interventions on back health should be longitudinal and compulsory in the curriculums during primary school.
Minor comments:
- Change Randomized control trials to ‘Randomized Controlled Trials’
Author’s response: The change has been checked and made to the entire manuscript
- I would recommend the authors to have the manuscript checked by a native English speaker or a translator, or give the manuscript another careful read-through. I found non-academic language, an example: “Page 2, line 57: These talk about…”
Author’s response: Thank you very much for the grammar review and suggestion. The article was sent to review by a native English again.
- Table 4: What are the effect sizes reported in the table? Betas? Can you please add a legenda to this table. Please describe this in the text as well.
Author’s response: The studies did not provide effect size or beta. As you indicate, it is data that we would have liked to offer to study the effectiveness of the intervention.
Reviewer 3 Report
Thank you for the opportunity to review this manuscript
The aim of the present study was to identify, describe, and analyze the educative features of RCTs on back health developed to date in the school setting.
The most important aspect of this article is the need to increase knowledge and research on Postural Education in the school context.
This review will serve as a basis for future works.
This article presents a good level and I only suggest small modifications.
Line 141 Delete the space between by and students…
Line 141 Check the number 84 and delete or change the citation number
Line 152 Space is needed between 1 and hour….
Line 167 Delete the space between post-test, and and…
Line 170 Delete the space between the and control group…
Line 182 An endpoint is needed at the end of the paragraph.
Line 332 Delete the space between clarify and the role….
Line 358 Delete the space between clarify and the role….
Line 363 Delete ….5. Conclusions…( at the end of the paragraph)…and delete lines number 364, 365, 366, 367 … seem to belong to another article
Author Response
Responses to Reviewers' Comments
[IJERPH] Manuscript ID: ijerph-1043243
We thank the reviewers for their thoughtful and in-depth comments concerning our manuscript. Your suggestions helped us improve the quality of our paper. We carefully considered every comment and made the appropriate changes, which are highlighted in red font. Besides the manuscript has been edited by a specialist native speaker. Our point-by-point responses are also noted below.
Reviewer #3:
Thank you for the opportunity to review this manuscript
The aim of the present study was to identify, describe, and analyze the educative features of RCTs on back health developed to date in the school setting.
The most important aspect of this article is the need to increase knowledge and research on Postural Education in the school context.
This review will serve as a basis for future works.
This article presents a good level and I only suggest small modifications.
Comments:
Line 141 Delete the space between by and students…
Author’s response: All the spaces between words have been revised and we believe that now they are all corrected
Line 141 Check the number 84 and delete or change the citation number
Author’s response: In order not to confuse the reader, the phrase has been corrected
“and reporting Cronbach’s alpha (α= .84)”
Line 152 Space is needed between 1 and hour….
Author’s response: It has been corrected. from 1 hour to 7 lessons
Line 167 Delete the space between post-test, and and…
Author’s response: It has been also corrected post-test, and
Line 170 Delete the space between the and control group…
Author’s response: It has been also corrected the control group
Line 182 An endpoint is needed at the end of the paragraph.
Author’s response: It has been corrected
Line 332 Delete the space between clarify and the role….
Author’s response: It has been corrected clarify the role of formal
Line 358 Delete the space between clarify and the role….
Author’s response: It has benn corrected
Line 363 Delete ….5. Conclusions…( at the end of the paragraph)…and delete lines number 364, 365, 366, 367 … seem to belong to another article
Author’s response: Sorry for the mistake. Has been corrected and removed. Thank you very much again.
Round 2
Reviewer 1 Report
It seems that it has been improved.Reviewer 2 Report
Dear Authors,
I do not have any further comments, and am happy with the way you addressed the issues that were raised.